# Visual Neurorestoration: An Expert Review of Current Strategies for Restoring Vision in Humans

**DOI:** 10.3390/brainsci15111170

**Published:** 2025-10-30

**Authors:** Jonathon Cavaleri, Michelle Lin, Kevin Wu, Zachary Gilbert, Connie Huang, Yu Tung Lo, Vahini Garimella, Jonathan C. Dallas, Robert G. Briggs, Austin J. Borja, Jae Eun Lee, Patrick R. Ng, Kimberly K. Gokoffski, Darrin J. Lee

**Affiliations:** 1Department of Neurological Surgery, Keck School of Medicine, University of Southern California, Los Angeles, CA 900033, USA; jonathon.m.cavaleri@gmail.com (J.C.); michelle.lin@med.usc.edu (M.L.); kevin.wu@usc.edu (K.W.); jacklo@nus.edu.sg (Y.T.L.); jonathan.dallas@med.usc.edu (J.C.D.); robert.briggs@med.usc.edu (R.G.B.); austin.borja@med.usc.edu (A.J.B.); jaeeun.lee@med.usc.edu (J.E.L.); patrick.ng@med.usc.edu (P.R.N.); 2Department of Ophthalmology, San Antonio Military Medical Center, Lackland, TX 78236, USA; zgilbert@usc.edu; 3Department of Ophthalmology, Keck School of Medicine, University of Southern California, Los Angeles, CA 90033, USA; huangcon@usc.edu (C.H.); vgarimel@usc.edu (V.G.); kimberly.gokoffski@med.usc.edu (K.K.G.); 4Department of Biomedical Engineering, Viterbi School of Engineering, University of Southern California, Los Angeles, CA 90033, USA; 5Department of Neurosurgery, National Neuroscience Institute, Singapore 308433, Singapore; 6USC Neurorestoration Center, Keck School of Medicine, University of Southern California, Los Angeles, CA 90033, USA

**Keywords:** visual impairment, blindness, genetic engineering, stem cell therapy, optogenetics, neuromodulation, eye transplant

## Abstract

Visual impairment impacts nearly half a billion people globally. Corrective glasses, artificial lens replacement, and medical management have markedly improved the management of diseases inherent to the eye, such as refractive errors, cataracts, and glaucoma. However, therapeutic strategies for retinopathies, optic nerve damage, and distal optic pathways remain limited. The complex optic apparatus comprises multiple neural structures that transmit information from the retina to the diencephalon to the cortex. Over the last few decades, innovations have emerged to address the loss of function at each step of this pathway. Given the retina’s lack of regenerative potential, novel treatment options have focused on replacing lost retinal cell types through cellular replacement with stem cells, restoring lost gene function with genetic engineering, and imparting new light sensation capabilities with optogenetics. Additionally, retinal neuroprosthetics have shown efficacy in restoring functional vision, and neuroprosthetic devices targeting the optic nerve, thalamus, and cortex are in early stages of development. Non-invasive neuromodulation has also shown some promise in modulating the visual cortex. Recently, the first in-human whole-eye transplant was performed. While functional vision was not restored, the feasibility of such a transplant with viable tissue graft at one year was demonstrated. Subsequent studies are now focused on guidance cues for axonal regeneration past the graft site to reach the lateral geniculate nucleus. Although the methods discussed above have shown promise individually, improvements in vision have been modest at best. Achieving the goal of restoration of functional vision will clearly require further development of cellular therapies, genetic engineering, transplantation, and neuromodulation. A concerted multidisciplinary effort involving scientists, engineers, ophthalmologists, neurosurgeons, and reconstructive surgeons will be necessary to restore vision for patients with vision loss from these challenging pathologies. In this expert review article, we describe the current literature in visual neurorestoration with respect to cellular therapeutics, genetic therapies, optogenetics, neuroprosthetics, non-invasive neuromodulation, and whole-eye transplant.

## 1. Introduction

Visual impairment is a major source of disability globally, with approximately 188.5 million people suffering from mild visual impairment, 216.6 million people living with moderate to severe visual impairment, and 36 million people who are blind [1]. The prevalence of age-related complete blindness has been decreasing globally, but the prevalence of moderate to severe visual impairment has not [2]. In the United States (US) alone, 7.08 million people are living with visual impairment, of which 1.08 million are blind. Beyond the profound impact of vision loss on functional status and quality of life, visual impairment is also associated with a significant economic loss. Globally, vision loss is estimated to cost USD 411 billion in lost productivity [3]. In the US, the estimated economic burden of direct and indirect costs of vision loss is USD 134 billion [4]. Given this high prevalence and economic impact, there is a clear need for further strategies to combat vision loss.

The leading global causes of moderate to severe visual impairment are uncorrected refractive errors (86.1 million cases) and cataracts (78.8 million cases) [2]. The leading causes of blindness are cataracts (15.2 million cases), glaucoma (3.6 million cases), uncorrected refractive error (2.3 million cases), age-related macular degeneration (AMD) (1.8 million cases), and diabetic retinopathy (0.81 million cases) [2]. In the US, the leading cause of blindness in white Americans is AMD (54%), followed by cataracts (9%). Cataracts were the leading cause of blindness in black Americans (37%), followed by glaucoma (26%). Meanwhile, glaucoma and cataracts were tied (28.6%) for the leading cause of blindness among Hispanic Americans [5]. Correction of refractive error and cataract surgery can markedly reverse vision loss, but AMD, glaucoma, and diabetic retinopathy often result in irreversible vision loss, necessitating novel forms of neurorestoration.

There are a variety of interventions in the armamentarium for vision correction in acquired diseases. Correction of refractive error with eyeglasses dates back centuries and is highly effective [6]. Surgical removal of cataracts is dramatically effective for restoration of visual function [7]. For glaucoma, the mainstays in treatment target decreasing aqueous humor production or facilitate outflow to decrease intraocular pressure with pharmacologic agents (such as prostaglandin analogs, beta-blockers, alpha-agonists, and carbonic anhydrase inhibitors), laser treatments, or surgical shunting procedures [8]. Treatments for neovascular or wet AMD (nAMD) and diabetic retinopathy are similar in that both utilize intraocular administration of vascular endothelial growth factor (VEGF) inhibitors, laser therapy to curtail the growth of aberrant blood vessels, and vitrectomy for scar and hemorrhage removal [9]. For glaucoma, AMD, and diabetic retinopathy, treatments are mostly designed to prevent further damage; however, novel regenerative techniques will be required to reverse damage caused by chronic disease. Similarly, congenital causes of visual impairment result in irreversibly lost function in tissues in the visual axis. Inherited retinal disorders are a broad, genetically diverse subset of genetic disorders that cause visual impairment and blindness [10]. Treatment of these disorders is challenging since the light-sensing apparatus is often missing altogether, necessitating regenerative or neuromodulatory therapies [11].

Given these challenges in visual restoration, researchers and clinicians have had to develop creative and novel strategies. Advances in stem cell biology and genetic engineering have already been applied to patients with vision loss, and the horizon is brimming with further exciting developments [12]. Combining genetic engineering strategies with optogenetic techniques, several groups have applied optogenetic systems to patients with partial restoration of vision [13]. Innovations in surgical technique have also demonstrated the feasibility of whole-eye transplant with an ultimate goal of restoration of visual function [14]. Neuroprosthetics provide an alternate strategy in which electronic stimulation is applied directly to the neural elements to bypass damaged tissues to restore lost function [15]. These neuroprosthetics have already been successfully utilized in the form of retinal implants [16], optic nerve implants [17], and cortical implants [18]. Finally, non-invasive neuromodulation has shown early promise in improving vision [19]. The various modalities for visual neurorestoration with their visual outcomes are displayed in Table 1. These novel approaches for visual restoration are in their nascency, and no strategies have yet been able to fully restore vision, but the groundwork has been set for future innovations aimed at ultimately restoring sight to the blind.

There have been multiple comprehensive review articles recently published on stem cell therapy [20], gene therapy [12,21], optogenetics [13,22], whole-eye transplantation [14], neuroprosthetics [23], and neuromodulation [19]. However, much of this literature is targeted toward a singular audience, dominated by ophthalmologists and researchers in ophthalmology. This expert review article seeks to combine recent innovations from a large breadth of disciplines, ranging from cell biology to applied neurophysiology, into one article with a clinical focus. By doing so, we hope to expand the appeal of the subject to a broader clinical neuroscience audience. This expert review article seeks to answer several important questions that will guide future work in visual neurorestoration. First, what is the current state of research and clinical implementation of visual neurorestoration across multiple modalities? Secondly, what clinical outcomes have been achieved toward the goal of restoring sight? Lastly, what are the limitations and challenges facing the field, and what are some realistic ways that these may be overcome? The visual pathway is a complex system, and it will, therefore, require complex solutions to address malfunctioning components. Although all of the innovative approaches discussed in this manuscript have been the result of ingenious work, none of these strategies provides a singular answer to visual neurorestoration; therefore, further advances will be required, and successful visual neurorestoration will require the collaborative efforts of multiple disciplines.

## 2. Visual Pathway Anatomy

We will start our review with a brief examination of the anatomy of the visual pathway, as understanding the interventions requires some knowledge of the anatomical targets. The visual apparatus functions to focus incoming light onto the light-sensitive retina, converting this information into topographically oriented neural signals that travel via the optic nerve to the diencephalon and ultimately the cortex [24]. After light passes through the fibrous tunic of the eyeball, specific wavelengths of the electromagnetic spectrum are converted by the retina into neural signals [24]. These neural signals are subsequently transmitted through the visual pathway for processing and sensory integration, resulting in sight [24]. This highly specialized apparatus rapidly adapts to changes in the external environment, such as dimming of light, both by physically modulating the amount of light that passes through, as well as through alterations of photoreceptor responsiveness [24].

The outermost layer of the eye is composed of the fibrous tunic [25]. This fibrous layer is formed by white sclera around most of the eye, transitioning into the transparent cornea in the center. Extraocular muscles attach to the sclera, coordinating eye movement. The convex shape of the cornea facilitates light refraction, focusing light waves onto the retina [25]. The underlying vascular tunic, comprising the ciliary body, iris, and choroid, further focuses light through changes in the pupillary aperture and lens accommodation. Sympathetic and parasympathetic signals activate dilator pupillae and sphincter pupillae muscular contraction in the iris, respectively, altering pupillary size and ultimately the amount of light that is allowed to pass through [26]. Contraction of the ciliaris muscle slackens the suspensory ligaments, increasing the spherical shape of the lens. This process of lens accommodation allows for further refinement of light refraction, ensuring precise focusing of light signals on the perceptive region of the retina [26]. With age, loss of lens and ligamentous flexibility limit this accommodation, resulting in an inability to focus on near objects known as presbyopia [26]. Accumulation of crystalline proteins in lens fibers produces cataracts that also worsen visual acuity [27]. Replacement of the degenerative lens with an artificial intraocular lens reliably improves visual acuity, and newer lenses employ diffractive optics in an attempt to provide accommodation [7]. Aqueous humor is produced by the ciliary processes in the posterior chamber and transmitted to the anterior chamber (between the lens and cornea), where it is subsequently absorbed into the canal of Schlemm [28]. Obstruction of aqueous humor egress leads to an elevation of intraocular pressure and likely deformation of retinal ganglion cell (RGC) axons at the level of the lamina cribosa. Chronic elevation of intraocular pressure leads to RGC degeneration or glaucoma [29]. Medical management focuses on either decreasing aqueous humor production or promoting drainage through physiological pathways [29]. Chronic untreated glaucoma can lead to vision loss, and refractory disease may require surgical shunting of extra-anatomic pathways such as trabeculectomy [30].

After passing through the fibrous cornea and lens, light travels through the gelatinous vitreous body to the retina posteriorly [31]. Also known as the sensory tunic, the retina is responsible for phototransduction [24]. The outermost layer of the retina is Bruch’s membrane, which is in direct contact with the choroidal blood vessels [32]. Close to this is the outer retinal pigment epithelium (RPE); its melanin pigmentation plays an essential role in the prevention of light scattering. The RPE is also necessary for nutrient transportation from the choroid and phagocytosis of debris [32]. On the inner side of Bruch’s membrane sits the photoreceptor layer of rods and cones that convert electromagnetic light signals into electrochemical neural signals through phototransduction [33]. Light photons convert 11-cis retinal to all-trans retinal, releasing opsin from rhodopsin [34]. Through a multi-step signaling pathway, the release of opsin modulates G protein-coupled receptors, closing cation channels, with resultant cellular hyperpolarization [33]. The convergence of multiple rods on bipolar cells allows for scotopic vision, which amplifies dim light with limited resolution [35]. Conversely, when ample light is available, differential activation of red, green, and blue cones allows for trichromatic phototopic vision [33]. The highest concentration of cones is found at the center of vision, the macula and fovea, facilitating high-acuity color vision [24]. The cell bodies of these photoreceptors comprise the outer nuclear layer and traverse the external limiting membrane and outer nuclear layer to synapse with bipolar cells in the outer plexiform layer [24]. The cell bodies of bipolar cells, horizontal cells, and amacrine cells are situated within the inner nuclear layer, superficial to the outer plexiform layer [35]. Loss of photoreceptor inhibition on bipolar cells allows for propagation of glutamate signaling from bipolar cells to RGCs in the inner plexiform layer [35]. GABA inhibition from horizontal cells alters photoreceptor sensitivity as the availability of light fluctuates in our external environment; amacrine cells similarly modulate reactivity, but in RGCs instead [35]. The final layers of the retina are the ganglion cell layer formed by the RGC bodies, the retinal nerve fiber layer containing RGC axons, and the terminal internal limiting membrane [24]. RGC axons traverse toward the optic disc and make a right-angle exit through the eye at the optic disc, forming the optic nerve. A blind spot is found at the optic disc due to an absence of photoreceptors in this location [24]. Disparities between the distances where light is perceived on each retina facilitate binocular depth perception [36]. Inherited retinal disorders, direct traumatic injury, AMD, diabetic retinopathy, and glaucoma all contribute to loss of retinal cell layers with a resultant decrease in visual function [37]. A diagram of the eye and retinal cell layers is shown in Figure 1, which depicts the gross anatomy of the eye on the left-hand side and shows the path of light through the anterior structures to the retina. On the right-hand side of the figure is a zoomed-in cross-section that shows the cellular layers of the retina, with the side facing toward the vitreous at the top and the side facing the choroid at the bottom (figure created with BioRender.com).

RGC axons form the optic nerve, transmitting axon potentials through the optic chiasm and optic tracts, synapsing on the thalamic lateral geniculate nucleus (LGN) [24]. Nasal visual fields are projected onto the temporal hemiretina, and signals are subsequently carried in the optic nerve and down the ipsilateral optic tract laterally [24]. Inversely, temporal visual fields are perceived by the nasal hemiretina; signals start in the ipsilateral optic nerve but subsequently cross to the contralateral optic tract at the optic chiasm [24]. The LGN is made up of six layers [38]. Layers two, three, and five receive information from the ipsilateral temporal hemiretina, whereas layers one, four, and six form synaptic connections with the medial optic tract originating from the contralateral nasal hemiretina. Signaling between the LGN and midbrain coordinates saccadic eye movements at the superior colliculus and pupillary light reflex/vertical gaze at the pretectal nucleus. Compression, trauma, or ischemia along these pathways results in deficits of the corollary visual field quadrant. This damage can induce proximal Wallerian degeneration over time and RGC loss, which is an important consideration in addressing pathologies related to the optic nerve and RGC damage.

The next step of visual processing involves optic radiation (geniculocalcarine tract), which is the projection of visual fibers from the diencephalon to the cortex [24]. First, fibers are projected from the LGN to the primary visual cortex (striate cortex) in the posterior occipital lobe through one of two pathways. Fibers from the upper retinal quadrants (inferior visual field) travel through the inferior retinal fibers in the temporal lobe, also known as Meyer’s loop, to the inferior calcarine fissure (lingual gyrus) of the occipital lobe. Fibers from the lower retinal quadrants (superior visual field) travel through the superior retinal fibers in the parietal lobe to the superior aspect of the calcarine fissure (cuneus). From the primary visual cortex, the dorsal stream responsible for spatial integration of location and motion projects to the prestriatal secondary visual cortex and posterior parietal visual association cortex. The ventral stream originates in the primary visual cortex and projects to the inferotemporal cortex, providing information on the color and shape characteristics of objects.

Given the intricacies of the optic apparatus and the delicate nature of these structures, novel strategies will be necessary to restore lost signals at any step in this complex pathway. Therapies are necessary for the optimization of physical properties, such as loss of structural integrity, replacement of degenerated signaling cells, and neuromodulation of cortical visual perception. Creative strategies are necessary to address each of the divergent mechanisms in each of these potential etiologies of vision loss.

## 3. Regenerative Therapies

Inherited and acquired eye diseases often result in loss of critical functions in the optic apparatus, and these disease processes primarily manifest at the level of the retina and optic nerve. This can occur in the form of loss of critical cellular function or loss of the neurosensory cells themselves. A considerable amount of work has gone into developing novel regenerative therapies to restore function, including cellular therapeutics, genetic engineering strategies, optogenetics, and tissue transplantation, which will be discussed in this section.

### 3.1. Cellular Therapeutics

One of the major challenges in restoring vision is that many pathologies result in irreversible damage and loss of retinal cells. The regenerative capacity of the retina is very limited, necessitating tissue or cellular replacement. Autologous translocation of retinal pigment epithelium, Bruch membrane, and choroid has shown successful engraftment with some evidence of improvement in vision for patients with geographic atrophy and nAMD [39,40,41]. Similarly, autologous transplantation of RPE from one eye to the other eye has been accomplished with improvement of vision in patients with nAMD [42]. These studies demonstrate the feasibility and potential efficacy of cell transplantation in visual restoration.

There have been significant advancements in stem cell biology in recent years [43]. These innovations have been applied to retinal stem cell replacement approaches [44]. Early work has shown that human embryonic stem cells (hESCs) can be differentiated into retinal-like progenitor cells, and researchers are developing techniques to successfully integrate these cells into the retina [45,46]. Further studies have shown that functional RPE cells and photoreceptors can be generated from hESCs [47,48,49]. This was similarly demonstrated with induced pluripotent stem cells (iPSCs) [50,51,52]. More recently, groups have utilized three-dimensional culture techniques to generate the optic cup in vitro, allowing for isolation of stem cells at higher purity with appropriate preserved cytoarchitecture [53,54,55].

RGCs are lost in a variety of retinal and optic nerve pathologies, so replacement of these cells will be essential for vision restoration [37]. Replacement of RCGs is particularly challenging given their unique cytoarchitecture and the need for robust axonal outgrowth, necessitating stem cell-based strategies for cellular replacement [56]. Several groups have developed techniques to generate RGCs from hESCs and iPSCs [56,57]. Further work has shown that with BDNF supplementation in vitro, iPSC-derived RGCs can produce functional axons [58]. A CRISPR-engineered fluorescent protein RCG reporter line has been developed. This fluorescent tag allows for purification through fluorescent-activated cell sorting, axon growth tracking on fluorescence microscopy, and a fluorescent readout with in vitro compound library screening [59]. While these innovations have set the stage for RGC cellular therapeutics, they have yet to be successfully employed clinically.

To date, there have been several in-human trials of stem cell-based therapies using hESCs. Schwartz et al. (2015) reported two phase 1/2 clinical trials wherein they performed subretinal transplantations of hESC-derived RPE on nine patients with AMD and nine patients with Stargardt’s macular dystrophy [60]. They demonstrated tolerability without any serious adverse outcomes. Over half of treated patients (10/18, 55.5%) experienced an improvement in best corrected visual acuity (BCVA), and only one patient had a decrement in BCVA [60,61]. Similarly, Song et al. (2015) transplanted hESC-derived RPE to two patients with dry AMD and two patients with Stargardt’s disease, with demonstrated safety and vision improvement in BCVA (improvement in 9–19 letters) for three out of four patients, without any significant adverse events[62]. In a long-term follow-up (3 years) of this cohort, Sung et al. (2021) found that one patient had an improvement in BCVA >5 points that was stable at 3 years, while the other patients had stable BCVAs [63]. In a phase 1 trial, da Cruz et al. (2018) delivered a hESC-derived sheet of RPE cells on a synthetic basement membrane to two patients with nAMD, and neither patient experienced any complications; both patches were viable on OCT and biomicroscopy at one year, and both experienced an improvement in BCVA of over 15 letters. [64]. A phase 1/2 clinical trial by Mehat and colleagues (2018) showed that hESC-derived RPE could be safely delivered to 12 patients with Stargardt’s disease without any evidence of uncontrolled proliferation or inflammatory complications. Despite a dose-dependent development of subretinal hyperpigmentation, no patients demonstrated sustained improvements in BCVA in the treated eye or significant improvements in quality of life reports (National Eye Institution Visual Function Questionnaire) [65]. In a phase I clinical trial by Li and colleagues (2021), hESC-derived RPE was delivered subretinally to seven patients with Stargardt’s disease, and they found stable BCVA in treated eyes at a 5-year follow-up [66]. Brant Fernandez et al. (2023) reported a phase I trial in which hESC-derived RPE cell suspensions were injected subretinally in 12 patients with Stargardt’s disease, and they found that there were no adverse events but no significant improvement in BCVA [67].

At our own institution, Kashani et al. (2018) generated RPE cells from hESCs and developed a method of placing a cell monolayer onto a novel parylene matrix [68]. This was implanted in a subretinal fashion in patients with dry AMD in a phase 1/2 trial [69]. The authors observed host photoreceptor integration on OCT with concurrent improvement in BCVA in one patient (17-letter improvement), improved visual fixation in two patients with no major complications, and no sign of rejection despite a donor/recipient mismatch in human leukocyte antigen (HLA) [70]. In a long-term follow-up (3 years) for this trial, Humayun and colleagues (2024) found that there were no severe adverse events, and implanted eyes were more likely to achieve a >5-letter BCVA (≤20/200) and less likely to have a >5 worsening of BCVA compared with non-implanted eyes [71].

There have also been a small number of in-human trials making use of iPSCs. Mandai and colleagues (2017) transplanted a sheet of fibroblast iPSC-derived RPE cells subretinally to a patient with nAMD, and they observed successful engraftment and stable vision in this case report [72]. In another study, Sugita et al. (2020) transplanted HLA-matched iPSC-derived RPE to five patients with nAMD, and despite an absence of treatment with systemic immunosuppressants, they found that transplanted cells were stable for up to one year [73]. One patient experienced retinal edema requiring epiretinal membrane removal, and mild inflammation/rejection was observed in two other patients.

In addition to trials with hESCs and iPSCs, there have been a small number of studies that have made use of autologous mesenchymal or central nervous system preparations. Park et al. (2014) delivered autologous bone-marrow-derived CD34+ cells in an intravitreal fashion to six patients with AMD, retinal vascular occlusion, or RP, and they found stable BCVA in all treated eyes at a six-month follow-up [74]. Oner and colleagues (2018) delivered autologous adipose-tissue-derived mesenchymal stem cells suprachoroidally to four patients with dry AMD and four patients with Stargardt’s disease, and at a six-month follow-up, they found an improvement in BCVA in all eight patients, with no significant safety concerns [75]. In another study investigating suprachoroidal implantation of autologous adipose-tissue-derived mesenchymal stem cells, Limoli et al. (2018) delivered cells to 11 patients with dry AMD, with 14 control patients, and they found an improvement in BCVA at a six-month follow-up [76]. In another study using intravitreal autologous bone-marrow-derived cells, Tuekprakhon and colleagues (2021) showed stable BCVA at a 12-month follow-up with one severe but manageable adverse event [77]. Finally, Nittala et al. (2021) delivered human central nervous system stem cells subretinally to 15 patients with geographic atrophy caused by nAMD, and they found that the progression of geographic atrophy was slowed in regions in which stem cells were injected, but the study was not powered to observe changes in BCVA [78].

Overall, stem cell-based therapies have been demonstrated to be safe with few reported adverse events, and many trials have shown a significant improvement in BCVA (>5 letters). More long-term data and more patients will be necessary to demonstrate the clinical efficacy of these therapies, but these clinical trials provide reason for excitement. A diagram that depicts the basic concepts in stem cell strategies is presented in Figure 2. This diagram shows cells that are harvested from iPSCs or hESCs and then differentiated into retinal stem cells, which can then further differentiate terminally into photoreceptors, bipolar cells, or RGCs. The figure was created with BioRender.com.

### 3.2. Genetic Engineering

Rather than supplying whole cells to replace the function of damaged or dysfunctional cells, as in the cellular therapies described above, genetic engineering strategies aim to replace aberrant gene function in visual pathologies. With recent advancements in genome editing, innovative genetic engineering strategies targeting a variety of etiologies of vision loss have been developed [21]. Genetic engineering refers to a process by which gene-encoding DNA sequences are vector-delivered to tissues with the goal of restoring lost gene function, providing a new functionality, or interfering with aberrant gene functioning [79]. CRISPR-Cas9 is the primary method by which gene editing is performed in the modern era [80,81,82,83], and transgenes are typically packaged into adenoviral (AV) or adeno-associated viral (AAV) vectors for delivery to the target tissues [84]. Delivery is typically performed in either an intravitreal, subretinal, or suprachoroidal fashion [21]. Alternatively, some genetic engineering approaches make use of an ex vivo technique in which cells are harvested from another source, gene-edited, and then reintroduced into the target tissue [85]. A diagram that depicts genetic engineering strategies is presented in Figure 3, which illustrates how a transgenic element can be packaged along with CRISPR-Cas9 Machinery into a viral vector and then transfect the cell, integrate its transgenic element, and alter or restore gene expression within the retinal target cell. The figure was created with BioRender.com.

Inherited retinal diseases (IRDs) are a genetically and phenotypically diverse set of disorders affecting the retina for which genetic engineering offers promise. Leber congenital amaurosis (LCA) is the most common inherited cause of blindness in children, and homozygous loss-of-function mutations in RPE65 are one genetic cause of this disease. In a landmark trial, voretigene naparvovec-ryzyl (AAV2-hRPE65V2) was delivered to patients with RPE65-mediated inherited retinal dystrophy, which resulted in improved vision, leading to FDA approval of this gene therapy [86]. Trials are ongoing for other IRDs, such as achromatopsia [87,88,89,90,91], Bietti’s crystalline dystrophy [92,93], choroideremia [94,95,96,97,98,99,100], autosomal and X-linked retinitis pigmentosa (RP) [101,102,103,104,105,106], Stargardt disease [107], Usher syndrome [108], X-linked retinoschisis [109,110,111,112], and Leber Hereditary Optic Neuropathy (LHON) [113,114,115,116,117,118,119,120,121,122,123,124,125,126,127].

Genetic engineering strategies are not just limited to monogenic conditions. There are exciting applications in common acquired disorders such as diabetic retinopathy and nAMD [21]. Vascular endothelial growth factor (VEGF) signaling plays an important role in the pathophysiology of nAMD and diabetic retinopathy [128], and current treatments of these conditions, such as intravitreal injection of anti-VEGF monoclonal antibodies, focus on targeting this pathway [129]. Initial gene therapy trials have made use of rAAV-sFLT-1, a VEGF inhibitory protein carried by the AAV vector. Rakoczy et al. (2015) injected six patients with nAMD and found at a one-year follow-up that four of the six did not require rescue anti-VEGF, while two required one rescue injection, and there were no significant adverse events [130]. Subsequent studies found that rAAV-sFLT-1 was well tolerated with few adverse events, but there was no statistically significant difference in BCVA or retreatments with anti-VEGF antibodies compared to the control subjects [131,132]. RGX-314 is a gene therapy that uses AAV8 to deliver a transgene encoding a monoclonal anti-VEGF antibody fragment. In a phase 1/2a dose-escalation study, Campochiaro et al. (2024) found that injecting 6 × 10^10^ copies resulted in stable RGX-314 expression with stable to improved BCVA and no need for repeat anti-VEGF injections [133]. RGX-314 is currently in phase 2 and 3 clinical trials for the treatment of nAMD (NCT04704921 [134]; NCT04514653 [135]) and diabetic retinopathy (NCT04567550 [136]). These trials have demonstrated efficacy and durability with a decreased need for anti-VEGF antibody injections. Intravitreal anti-VEGF treatments are expensive and require access to extensive healthcare infrastructure, resulting in unsurprising access disparities [137]. A one-time delivery of gene therapy could potentially ameliorate some of the disparities related to healthcare access.

Other diseases, such as non-exudative AMD and glaucoma, may also be amenable to genetic engineering therapies. Non-exudative AMD is a common disorder for which there are currently limited treatment options. The complement cascade has been implicated in the pathophysiology of non-exudative AMD [138]. Trials using a vector encoding a complement inhibitor, Complement Factor 1, were terminated due to futility [139], but another gene therapy (JNJ-81201887) inhibiting the complement cascade is showing considerable promise [140]. JNJ-81201887 is an AAV2 vector carrying sCD59 that upregulates CD59 expression in RPE cells and has been shown to slow the progression of lesion growth in non-exudative AMD [140]. In a phase 1, first-in-human clinical trial by Heier and colleagues (2024), 17 patients with AMD received a single intravitreal injection of a viral vector expressing soluble CD59. No serious adverse events were reported, though some patients (29%) experienced mild ocular inflammation, which was resolved with topical steroids (80%). Patients served as their own controls; in the treated eye, geographic atrophy growth rate demonstrated a reduction from 0.211 to 0.056 mm over the 2-year period [140]. Given the success of this therapy, it has been fast-tracked by the FDA [141]. So far, there have been no in-human trials of gene therapies for glaucoma, but there are promising animal studies—all with the goal of lowering intraocular pressure (IOP)—by targeting prostaglandin biosynthesis in the anterior chamber [142], aquaporin 1 function in the ciliary body [143], beta adrenergic receptor function [144], matrix metalloproteinases [145], myocilin [146], and transforming growth factor beta [147].

As technologies improve, some of the limitations of gene therapy will be overcome, and these treatments will become increasingly more effective. There are several major issues with current gene therapies for retinal disorders. First, even though the eye is reasonably immune-privileged, the viral capsids still activate the innate and adaptive immune systems, which can damage the sensitive structure of the retina. Repeat administrations of gene therapy may also be quickly identified by immunologic memory mechanisms [148,149]. Second, the viral vectors have limited capacity for larger transgenes [150]. Third, a potential risk of subretinal delivery is retinal detachment [151], damage to the optic nerve [151], and macular holes [152], but subretinal delivery has higher transduction efficiency than intravitreous injection [153]. Novel techniques like non-invasive suprachoroidal injections of DNA-containing nanoparticles may help mitigate some of these problems [154].

### 3.3. Optogenetics

Optogenetics is an alternate strategy of genetic engineering that seeks to restore light-sensitivity to the retina through transgenic proteins rather than the replacement of lost natural gene function. In recent years, exciting innovations in the field of optogenetics have been applied to visual restoration [22]. Optogenetics refers to a technique wherein light-sensitive ion channels change their permeability to selective ions, causing either depolarization or hyperpolarization of the cells in which they are expressed [155,156,157]. Genetic engineering techniques (such as CRISPR-Cas9) have been developed to deliver genes encoding these optogenetic proteins (via viral vectors) to target cells within the retina [158]. Most optogenetics platforms make use of microbial channelrhodopsin-2 (ChR2) from the green algae *Chlamydomonas reinhardtii* or genetically modified derivatives of this protein [159], although other opsins have also been developed [22]. Bi and colleagues (2006) were the first to report partial restoration of vision in a murine model of photoreceptor degeneration following ChR2 delivery (with an adenoviral vector) to RGCs [160].

The basic concept of optogenetics for visual restoration is to re-establish light sensitivity to the retina by expressing optogenetic proteins in retinal cells, as illustrated in Figure 4, which depicts the optogenetic packaging and delivery of the optogenetic machinery (ChR2) to the retina on the right side of the figure, and on the left side of the figure, the transmembrane optogenetic protein (ChR2) increases membrane permeability to cations causing depolarization of the plasma membrane. The figure was created with BioRender.com. Obviously, this strategy is dependent on some preservation of target cells and normal tissue architecture. RP is characterized by preferential loss of rods with relative preservation of the light-sensitivity-diminished cones; therefore, targeting the dormant cone cells with hyperpolarizing opsins is one potential therapeutic strategy [161]. If the photoreceptors are lost, bipolar cells present an alternative target for optogenetics [162]. RGCs typically survive longer than photoreceptors and bipolar cells, making them an attractive target for optogenetics [163].

There have been several in-human trials of optogenetics for visual restoration. In the landmark PIONEER study, Sahel et al. (2021) delivered ChrimsonR, a red-shifted ChR2 variant, via an AAV2 vector subretinally to RGCs in a blind patient with RP [164]. The patient experienced partial restoration of vision with the aid of specialized goggles and was able to perceive, count, locate, and touch objects presented on a table in front of him. Another phase I/II clinical trial is enrolling 12 patients with RP for subretinal delivery of an AAV2 vector carrying ChR2 (NCT02556736 [165]). Another phase 1/2 trial is underway for RP, in which ChronosFP (a modified microbial opsin) is being delivered via an AAV vector (NCT04278131 [166]). The RESTORE trial is a phase 2b study that is currently enrolling 18 subjects with RP who will receive AAV2-delivered MCO1 (a ChR2 variant) targeting bipolar cells (NCT04945772 [167]). Similarly, the STARLIGHT trial is an ongoing phase 2 trial in which 10 patients with Stargardt’s disease have received AAV2-delivered MCO1, targeted at bipolar cells (NCT05417126 [168]).

Optogenetic applications for visual restoration are still early in development, and surely more innovations will allow for improved treatment options. One of the main barriers is that current optogenetic tools are not ideal for natural light conditions, so optogenetic tools will need to be further optimized for ambient-light wavelengths [22]. More work will also be required to determine which optogenetic tools are best for different disease types.

### 3.4. Whole-Eye Transplant

Allogenic whole-eye transplantation is another form of regenerative therapy that seeks to replace the entire tissue rather than individual cells or gene products. Whole-eye transplantation is another theoretical alternative to tissue replacement, with prior precedent for this approach. Recently, Li and colleagues (2025) developed the first orthotopic rodent whole-eye transplantation model, demonstrating good long-term graft and animal survival [169]. In a landmark first-in-human study, Dr. Eduardo Rodiguez and colleagues (2025) successfully performed an allogenic partial face and whole-eye transplantation for a patient who had suffered severe burns [170]. The patient did not regain vision in the donor eye, but the donor eye tissue remained viable at 1-year follow-up. This monumental effort demonstrates the feasibility of whole-eye transplant, but some important barriers will need to be overcome before this becomes a viable clinical option. There are currently five groups that have been awarded ARPA-H Transplantation of Human Eye Allografts (THEA) grants with the goal of restoring vision through whole-eye transplant (InGel Therapeutics in Allston, MA; Stanford University; the University of Colorado Anschutz Medical Campus; and the University of Miami Bascom Palmer Eye Institute) [171].

Whole-eye transplant offers some advantages over cellular replacement, as the complex cytoarchitecture and neurovascular connections are preserved in the donor eye [14]. The principal limitation of whole-eye transplant is that donor RGC axons need to traverse the length of the recipient’s optic nerve, past the donor–recipient optic nerve interface. Several strategies have shown moderate efficacy in rodent optic nerve regeneration models, including PTEN and SOCS3 gene deletion [172]; supplementation with growth factors such as oncomodulin [173,174]; fibroblast growth factor 2 [175]; and constitutive activation of mammalian sterile 20-like kinase 3b [176].

Although not sufficient for independent axon regeneration, other factors have been shown to positively influence RGC survival and augmentation of axon regeneration. For example, overexpression of anti-apoptotic Bcl family proteins like BCL-2 and BCL-xL has been shown to promote RGC survival [177]. Administration of trophic factors like ciliary neurotrophic factor [178], brain-derived neurotrophic factor [179], neurotrophin-4/5 [180], nerve growth factor (NGF) [181], insulin-like growth factor 1 (IGF-1) [182], glial-derived neurotrophic factor (GDNF) [183], and neurturin [184] have also been shown to promote RGC survival.

Work by our own group in Dr. Gokoffski’s lab has shown that an exogenous directional electric field applied to the rat optic nerve after crush injury provides a strong guidance cue for axon regeneration down the optic nerve toward the diencephalon, resulting in partial visual restoration (as shown in Figure 5, which depicts the experimental setup in panels A and B, with panel C showing axon regeneration past the crush injury along the optic nerve treated with an electric field; figure reproduced with permission from Dr. Gokoffski) [185]. Importantly, long-term electrical stimulation was possible in awake-behaving rats and did not appear to cause distress, indicating that this therapy might be applied comfortably to human patients. Current work is ongoing to demonstrate whether electric fields can be applied to human subjects.

There are significant challenges that will need to be overcome for whole-eye transplantation to be a viable strategy for visual restoration. First, the retina is a highly vascularized tissue that quickly becomes ischemic, so a system of perfusing the eye ex vivo between harvest and transplantation needs to be developed [186]. Also related to blood flow, anterior segment ischemia is a known complication of strabismus surgery in which the extraocular muscles are disrupted, so transplant of the entire optic cone would be ideal to prevent ischemic complications, thus abrogating the ability of the donor to move the transplanted eye [187]. Additionally, as discussed above, axon growth and myelination along the new fibers are necessary for vision. Related to axon growth, ensuring targeting of new fibers to the correct diencephalic and eventually cortical targets is essential to maintaining the retinotopic orientation [14]. Lastly, although the eye is an immune-privileged site, local microglial activation and astrocytic scar formation are inhibitory toward axon regeneration [188,189]; furthermore, optimal immunosuppression regimens will need to be elucidated to balance the risks of adverse medication-related reactions with rejection.

## 4. Visual Prosthetics

Another approach for visual restoration that does not involve replacing cells, gene products, or tissues is the implantation of visual neuroprosthetics—implantable devices that act to make up for lost function of various components of the optic pathway. Commonly, these devices act at the retinal level, cortical level, midway at the optic nerve/tract, or at the thalamic level [23]. The concept underlying all implants is the transfer of visual information from an external sensor, typically a camera mounted on glasses, to the implant, which subsequently stimulates the target structure in a visuotopic manner. The various neuroprosthetic devices are shown below: Figure 6A: Argus II retinal implant device, Figure 6B: IRIS-II retinal implant device, Figure 6C: PRIMA retinal implant device, Figure 6D: Orion cortical implant device, and Figure 6E: Gennaris cortical implant device.

### 4.1. Retinal Implants

Retinal implants target the retinal ganglion layers. Stimulation of these layers results in variable visual percepts, known as phosphenes. Patients commonly report these sensations as feeling artificial, with interpretation of visual perception dependent upon associative learning. Argus II (Cortigent, Inc., Valencia, CA, USA, formerly Second Sight Medical Products, Inc., Sylmar, CA, USA), PRIMA (Science Corporation, Alameda, CA, USA), IRIS II (Science Corporation, Alameda, CA, USA), and Orion (Cortigent, Inc., Valencia, CA, USA) are among the most prominent retinal implants.

The Argus II (originally developed by Second Sight Medical Products, Inc., Sylmar, CA, USA) is the only retinal implant that has achieved FDA approval, with a humanitarian device exemption (HDE) granted in 2013 and the CE mark in European markets (2011) for specific treatment of RP [195]. The Argus II device restores visual perception by using a glasses-mounted camera that transmits the image data to an epiretinal electrode array. Stimulation through the electrode array aims to activate the surviving ganglion and bipolar cells of the macular retina. In a study by Humayun and colleagues (2012), in which 30 patients were implanted with the Argus II (45.6 total subject-years), patients saw improvements in object localization (96% of patients), motion discrimination (57%), and oriented gratings discrimination (23%) [196]. Thirty percent of patients had at least one serious adverse event (SAE), with 17 SAEs in total. In a follow-up study of the same cohort by Duncan et al. (2017), quality of life measures were evaluated (measured by VisQoL) in 30 patients with the Argus II [197]. Composite VisQoL did not show any changes; however, 3/6 domains (likelihood of injury, coping with life demands, and fulfillment of life roles) showed significant improvement. Two domains (organizing assistance and confidence in everyday activities) trended toward significance. No patients had baseline deficits in the last domain (ability to have friendships). In 2019, production of the Argus II was halted by the manufacturer, but production was set to be resumed by Cortigent, Inc. in 2023 [198]. However, Cortigent, Inc. shut down the production of the Argus II device, citing a small population of patients with RP [199].

Alternatively, the Intelligent Retinal Implant System (IRIS) II device is an epiretinal implant developed by Pixium Vision SA (Paris, France). Like the Argus II, the IRIS II has achieved CE approval in the European Union and is designed to treat patients with RP. In a 6-month post-implant assessment of efficacy and safety with 10 RP patients, the IRIS II device demonstrated significant improvements in square localization (−6.25 cm at 6 months, *p* = 0.02), direction of motion (−35° at 6 months, *p* = 0.0078), and picture recognition (+6.45% at 6 months, *p* = 0.0078). No patients had a measurable visual field at baseline; 8/10 patients had a measurable visual field at 3 months (median 50°); and 5/9 had a measurable field at 6 months (mean 119°) [191]. There were six SAEs that were corrected by surgical revision in the trial, but device production was halted in 2018 [200].

The Photovoltaic Retinal Implant (PRIMA) bionic vision system is another visual neuroprosthetic developed by Pixium Vision SA (and later acquired by Science Corporation, Alameda, CA, USA), aimed at restoring visual function through a subretinal approach. In the PRIMA system, electrodes implanted between the choroidal and photoreceptor layers of the macula stimulate the photoreceptor layer based on inputs from a glasses-mounted camera [192]. In its first-in-human trial, the PRIMA system was implanted in five patients with geographic atrophy from AMD. All five had light perception at the implant site (primary endpoint). One patient had the implant incorrectly inserted, and one died 18 months after surgery (study/device-unrelated). Of the remaining three patients, visual acuity matched the pixel size of the implant (~Snellen 20/438–20/565). At 48 months, visual acuity improved to 32 ETDRS letters above baseline (using zoom) [192]. There is currently a 38-patient study (NCT04676854) underway in Europe with the PRIMA device [201].

Overall, retinal implants have shown some promising results with vision restoration; however, devices like the Argus II and IRIS II have already been discontinued by the manufacturers. These devices are expensive, require technical surgeries for implantation, and are only available for small populations of patients. For retinal implants to achieve more success, they might instead be targeted at more common pathologies, such as nAMD, diabetic retinopathy, or glaucoma.

### 4.2. Optic Nerve Implants

A promising approach developed by the Tano group in Japan is direct stimulation of the optic nerve, entirely bypassing the site of retinal injury in RP. The implant, named Artificial Vision by Direct Optic Nerve Electrical stimulation (AV-DONE), involves the direct implantation of thin wire electrodes into the optic nerve disc through a three-port vitrectomy [17]. In total, 3–5 electrodes are implanted and subsequently stimulated under local anesthesia, resulting in the perception of phosphenes; this was initially trialed in RP patients with partial vision loss and most recently in RP patients with total vision loss. However, the limited number of electrodes impacts the resolution of the perceived image, which may explain why this technology has not been further developed.

### 4.3. Thalamic Implants

The thalamus plays an important role in the optic pathway, as it houses the lateral geniculate nucleus. Implantation of visual neuroprosthetics into the thalamus remains theoretical and has yet to be explored. Some work in non-human primates has shown that visual percepts can be elicited with microstimulation of the LGN [202]. Given the small size of the thalamus and its deep location, practical applications of thalamic stimulation are far from reality [203].

### 4.4. Cortical Implants

While placement of neuro-electronic interfaces onto the visual cortex requires a more invasive implantation, including a craniotomy and durotomy, direct stimulation of the visual cortex might provide vision for patients for whom the aforementioned neuroprosthetics may be ineffective. For instance, retinal implants like the Argus II and IRIS II stimulate the non-degenerated layers of the retina, and this is predicated on intervention prior to a critical level of disease progression. Additionally, injuries more distal in the visual pathway, such as those afflicting the optic chiasm secondary to suprasellar compression or the optic nerve secondary to glaucoma, may render retinal and optic nerve stimulation moot. Furthermore, cortical magnification in the primary visual cortex offers additional opportunities for maximizing the effective resolution of electrode-array stimulation. However, there is significant integration of neural signals by the time they reach the visual cortex.

Direct cortical stimulation to generate phosphenes traces its origins back to early work by Brindley [204,205] and Dobelle [206,207,208], who showed that electric stimulation of the occipital cortex could generate visual percepts in awake patients. Further work by Beauchamp and colleagues showed that dynamic stimulation of the visual cortex could elicit shape, but when multiple contacts were stimulated simultaneously, they blurred into large, indistinct phosphenes [209].

Building off these discoveries, neuroprosthetic devices have been developed. The manufacturers of the Argus II device (formerly Second Sight Medical Products, Inc., now Cortigent, Inc.) also developed the Orion device, a subdural cortical implant that directly stimulates the visual cortex. Similar to the Argus II, the device utilizes an external, glasses-mounted camera for visual data that is wirelessly relayed to the stimulatory array, relying on head movements for gaze control [210]. However, the implantation of this device requires a craniotomy and durotomy. Patients have reported phosphene production following implantation of the Orion device [211]. The Orion II device is now under development by Cortigent, Inc. [200].

The Orion device uses macroelectrode contacts, which have a larger volume of tissue activation, which makes it difficult to stimulate individual receptive fields in the visual cortex when compared to microelectrodes [212]. Schmidt and colleagues implanted the first long-term microelectrode array with 38 electrodes in a female participant with vision loss from glaucoma [213]. Fernandez and colleagues implanted a 96-microelectrode Neuroport Utah array (Backrock Microsystems, Salt Lake City, UT, USA) in a participant with complete blindness, and the participant was able to identify some letters and recognize object boundaries [212]. Although these microelectrode implant systems were for research purposes only, they set the groundwork for the application of microelectrode-based strategies for visual neurorestoration.

The Gennaris system (Monash Vision Group, Melbourne, Australia) is another alternative cortical stimulation system. Similar to the Orion, the Gennaris is a cortical implant that communicates wirelessly with an external camera [194]. Preliminary ovine studies demonstrated the Gennaris system to be biocompatible at 3 months (the length of the study) without cellular damage upon chronic stimulation. This device is awaiting the start of in-human trials.

Although work in neuroprosthetics has produced some impressive results and yielded innovative technologies, much more work is necessary for cortical stimulation to result in the restoration of functional vision. For example, bidirectional information flow and closed-loop stimulation may allow for optimization of stimulation parameters. Also, wireless technologies could allow for more portability of implants, thereby making more maneuverability possible in real-world settings for patients. Furthermore, artificial intelligence could help in interpreting signals and tailoring stimulation programs to the visual cues. Undoubtedly, there will be more innovations in this realm in the years to come.

## 5. Non-Invasive Neuromodulation

In addition to the invasive neuromodulatory techniques described above, there have also been significant developments in non-invasive neuromodulation for visual neurorestoration. Technologies like repetitive transcranial magnetic stimulation (rTMS), transcranial direct current stimulation (tDCS), transcranial alternating current stimulation (tACS), and transcranial high-frequency random noise stimulation (tRNS) have shown some utility in visual restoration. Figure 6F depicts rTMS, but all non-invasive brain stimulation modalities have similar designs, in which an external element is applied to the scalp to stimulate the cortical surface.

Silvanus P. Thompson first experienced visual phosphenes in 1910 when his head was in close proximity to a strong magnetic field [154]. Since that time, the influence of magnetic fields on the brain has been harnessed into rTMS, which has been widely adopted in the treatment of psychiatric disorders [214]. rTMS makes use of coils of wire through which current is passed to generate a directional magnetic field, which can induce an electric field in the brain, thereby causing a physiologic effect [215]. rTMS for the generation of phosphenes is of limited clinical utility because it activates too many receptive fields in the visual cortex to allow for specificity, and it can also be inhibitory to visual processing [216].

The main clinical utility of rTMS is that it can induce plasticity in neural circuits through long-term potentiation [215]. rTMS applied to the visual cortex has been shown to improve contrast sensitivity temporarily in adult patients with amblyopia [217], as has continuous theta burst stimulation (cTBS), an alternate pattern of stimulation that mimics natural brain rhythms [218,219].

tDCS applies an external direct current to the brain, causing sustained cortical excitability [220]. Similarly, tACS applies a sinusoidal alternating current to the brain and modulates brain function [221]. tRNS applies random high-frequency noise, which has been shown to increase brain excitability [222]. All three modalities make use of an external device to supply current transcranially via an electric potential across the brain [223]. tDCS has been shown to improve stereopsis [224], as well as contrast sensitivity and visually evoked potential amplitude in patients with amblyopia [225]. When applied to the visual cortex in patients with hemifield deficits after stroke, tDCS was shown to improve motion perception [226] and visual field function [227]. A recent meta-analysis by Bello and colleagues investigated the pooled results from studies from tDCS, tACS, and tRNS (termed transcranial electric stimulation, tES) applied to patients with normal vision, showing that tES improves contrast sensitivity, increases visual evoked potential amplitudes, and reduces crowding in peripheral vision [228].

One potential application for non-invasive neuromodulation is restoring alpha (8–14 Hz) oscillations in the occipital cortex in patients with visual impairments and congenital blindness [229]. Patients with congenital blindness have decreased alpha activity in the visual cortex [229,230]. In patients with congenital cataracts, delays in surgical treatment result in worse visual outcomes due to abnormalities of visual cortex activity [231]. tACS has been shown to increase alpha activity in healthy subjects [232,233]. In a randomized trial, Middag-van Spanje and colleagues (2024) provided tACS at alpha frequencies to the visual cortex of patients with chronic visuospatial neglect after stroke in combination with visual scanning training, which resulted in improvements in neglect [234].

Non-invasive neuromodulation is a low-risk intervention and will undoubtedly have an expanded indication in the treatment of all neurologic disorders in the near future, including visual neurorestoration.

## 6. Challenges and Future Directions

A considerable degree of elegant and ingenious work has gone into developing regenerative and neuromodulatory therapies for visual neurorestoration; however, the overall improvements in vision have been modest with these therapies individually. Certain pathologies respond better to different methodologies, but the fact remains that the field is a long way away from the goal of restoring sight to the blind. Current and future developments are aimed at addressing many of the major issues with therapies across a broad spectrum of fields with the goal of making neurorestoration a more feasible reality.

In the field of cellular therapeutics, active research is underway to address some of the major problems facing the successful implementation of these treatments. One major barrier is immune rejection, which is primarily a concern with allogenic transplantation in which HLA mismatch is present [235]. Common immunosuppressants such as tacrolimus, mycophenolate, and steroids have been used in these cases with some success [236]. One strategy that could circumvent the problem of HLA mismatch is to have a bank of iPSC-derived stem cells with different HLA alleles from which matched cells could be selected for a patient [237]. Alternatively, the problem of rejection has largely been abrogated with the use of autologous cell transplants, such as iPSC-derived therapies. With iPSC-derived therapies, there are still considerable barriers to widespread implementation, as harvesting and processing are incredibly time- and resource-intensive [238]. Future work should be directed toward decreasing the costs and time of generating iPSC-derived stem cells. Use of small-molecule inhibitors and growth factors has already cut down the time and costs of generating stem cells compared to spontaneous development, but these methods need to be perfected and more widely implemented [239]. One potential strategy that has been investigated is in vivo differentiation, in which local cells can be reprogrammed to stem cells in their tissue of origin, which has been shown with BMP and sFRP2 inhibition in a murine model [240]. This could eliminate some of the costs and reduce the time of generating stem cells in vivo, but its effectiveness has yet to be demonstrated.

The next major frontier in cellular therapeutics is the successful implementation of an RGC-based therapy. RGCs have been shown to engraft in a murine retina and form some short-range synapses [241], but these cells did not show the long-range axon regeneration down the optic nerve that would be necessary for restoration of vision. One significant barrier to RGC axon regrowth is the lamina cribrosa, a collection of perforated fibroelastic plates through which RGC axons pass before converging into the optic nerve [242]. Guiding new axons through this structure poses a challenge, especially in diseases like glaucoma, which cause pathologic remodeling of the extracellular matrix of the lamina [243]. More work will be necessary to understand the guidance cues for axons traversing this structure. Strategies inhibiting PTEN, SOCS3, STAT3, Kruppel-like factors 4 and 9, and others could also be employed further to stimulate axon regeneration [244]. Furthermore, the optimal extracellular environment for axon regeneration in terms of trophic factors, inflammatory products, and glial components will need to be elucidated. Despite advances in the understanding and production of RGCs, more work is necessary to demonstrate that RGCs can be successfully recapitulated, and it is unknown whether they will be able to make retinotopic connections in the LGN and whether afferents from the LGN to the visual cortex would accurately convey this information to the proper receptive field. Successful RGC replacement strategies face considerable obstacles to implementation.

Similar to cellular therapeutics, gene therapy strategies also face a number of obstacles. One of the greatest barriers to successful gene therapy approaches is the immunogenicity of viral vectors. Although the eye is relatively sequestered from an immunologic perspective, interactions with the immune system can cause patient harm and can lead to diminished expression of the transgene [245]. A very high percentage of patients express neutralizing antibodies to previously encountered AAVs that have cross-reactivity to the commonly used AAV vectors (AAV1, AAV2, AAV8), which interferes with patients’ candidacy for gene therapy [246]. Treatment with immunosuppressants has been shown to decrease inflammation [247]. The route of injection and the AAV serotype have also been shown to modulate immunogenicity [248]. Current genetic engineering strategies (such as directed evolution and machine learning-guided design) have been harnessed to create AAV vectors that are safer and more potent [249]. Other non-viral vectors, such as lipid nanoparticles [250,251], are less immunogenic than viral vectors, but their main limitation is that the duration of the expression of their transgenes is generally short. The use of improved viral and non-viral vectors in future trials will hopefully improve the safety and efficacy of gene therapies.

As optogenetic strategies use viral vectors and genetic engineering technologies, the same limitations apply to optogenetics as well; however, there are some unique challenges to optogenetics that will need to be addressed in the future. Current optogenetics approaches make use of light-amplifying goggles that emit short-bandwidth light at high intensities. These goggles are expensive, bulky, and difficult to use in everyday situations, and there is concern about phototoxicity to the retina [13]. Strategies employing opsins that are sensitive to naturalistic light are, therefore, more appealing options. For example, newer opsins like ChRmine [252], ComV1 [253], and GHCR [254] have better responsiveness to natural ambient lighting conditions. Even with optimal technologies, transfection efficiency, stability of expression of optogenetic constructs, and the degree of damage to the cytoarchitecture of the diseased retina can lead to variability [13]. Future work will need to focus on improving the stability of expression of chromophores that are responsive to natural light and can be optimized for different pathologies.

In terms of future directions for whole-eye transplant, there are some challenges that will need to be overcome before this becomes an effective strategy for visual neurorestoration. First, successful optic nerve coaptation is a major barrier [14]. Donor selection, organ harvesting, and preservation of the highly ischemia-sensitive tissues through media selection and ex vivo perfusion techniques (all termed TA1 in the ARPA-H THEA grant) will need to be optimized first [171]. If the tissue can be successfully preserved, obtaining RGCs in the donor optic nerve to migrate past the coaptation site across the optic chiasm and tract to synapse with the LGN (all termed TA2) is another barrier [171]. Various groups are studying microsurgical anastomosis, gene therapies, growth factor supplementation, scaffold placement, and bioelectric techniques. Another challenge that will need to be overcome is the preservation of naturalistic eye movement. As mentioned earlier, transplantation of the optic cup (including the extraocular muscles from the donor) may help with perfusion of the anterior chamber [187], but this would require additional functional cranial nerve anastomosis (cranial nerves III, IV, and VI). These nerves are very small and sensitive, and there is no precedent in the literature for repairing damaged nerves to restore ocular movement. If the recipient’s extraocular muscles were instead preserved, this could allow for preservation of conjugate extraocular motion, but the problem of anterior chamber ischemia would have to be addressed. Rejection is another major concern that will need to be addressed. HLA matching should be attempted, and optimal immunosuppressant medication regimens will need to be elucidated that build on decades of research in solid-organ transplantation [255]. Even with successful optic nerve regeneration, considerable barriers remain. Future work will be concerned with guiding axons from the optic nerve to their proper ipsilateral and contralateral targets within the diencephalon with proper cues to preserve retinotopic architecture.

With regard to future directions in neuroprosthetics, these technologies are still a long way off from restoring functional vision. Although it is impressive that retinal implants have restored some visual function in patients who had no visual function whatsoever, they are far from achieving functional vision. There are several challenges that must be overcome for these technologies to be more widely adopted. As in the case of the Argus II, high currents are required, requiring larger electrodes, which limits the number and spacing of electrodes [256]. Future implants will employ technologies with higher-density arrays that allow for higher retinotopic resolution. Artificial intelligence strategies may also be employed to more accurately recapitulate naturalistic vision [257]. Other challenges to retinal implants are that these technologies are very expensive, the surgeries are complex and require expert surgical teams for implantation, there are small numbers of eligible patients, and the postoperative and rehabilitative care requires considerable time and support [256]. Future retinal implant technologies will need to be more generalized so that they can be given to patients with different pathologies of differing severity, and costs to patients will need to be decreased to improve access. Optic nerve and thalamic-based neuroprosthetics are still a long way off from achieving clinical utility [17,203]. Part of the issue is that these structures are small, sensitive, and difficult to access surgically. It is hard to imagine a scenario in which these approaches will be clinically useful, but they would certainly require the development of smaller, higher-density arrays that could provide more specific stimulation of these small structures.

Cortical-based neuroprosthetics are a promising strategy in visual restoration, as they can be applied to any ocular pathology as long as the visual cortex is healthy. Devices like the Orion and Gennaris system make use of macroelectrode contacts on the cortical surface [194,211]. Macroelectrode arrays are an appealing strategy as they are easy to implant and do not require violation of the pial surface, but their larger size stimulates multiple neurons, making the phosphenes generated less distinct [258]. Microelectrode arrays, on the other hand, are able to stimulate small groups of neurons or individual neurons, which dramatically increases the specificity of signals [213]. It is estimated that approximately 700 electrodes in a 1 × 1 cm grid would be necessary to recapitulate foveal vision [259], so even more electrodes would be necessary to restore full retinotopic vision. Future studies should be concerned with placing enough arrays to cover a large area of the visual cortex. For example, we envision a study in which multiple arrays (such as Utah Microelectrode arrays) will be placed. Having this many arrays would create a large parameter space for stimulation optimization, which would necessitate formidable computational power that would make instantaneous vision more difficult. This could be aided by deep learning and artificial intelligence techniques and more compact, powerful computer processors [260].

Non-invasive neuromodulation has shown some ability to improve vision, but there is still some work that needs to be done to improve upon these methods and to discover new applications in vision restoration. First, strategies like in-home therapies may decrease the cost barriers to receiving these therapies [261]. Secondly, different patients and pathologies may require different stimulation parameters, and discovering the optimal stimulation settings will be necessary to achieve maximal therapeutic effects [234]. Parameter selection could be aided by electrophysiology [262], neuroimaging [263], and computer modeling [264]. Non-invasive stimulation certainly does have some potential in vision restoration; however, many pathologies that cause vision loss would not be responsive to non-invasive stimulation. Therefore, the role of non-invasive brain stimulation is limited to a narrow spectrum of etiologies of vision loss.

Combining some different therapies that act at different levels of the visual axis may provide the best chance of visual restoration. For example, in a patient with glaucoma who has experienced chronic optic nerve degeneration and loss of RGCs, stem cell therapies could be used to restore RGCs, along with methods for axon regeneration such as PTEN inhibition, supply of growth factors like oncomodulin, and an implant to generate an electric field from the proximal optic nerve to the LGN. Alternatively, in a patient with an inherited retinal disorder such as RP with degenerated photoreceptors, an optogenetic platform like the ChrimsonR-based system used in the PIONEER study could be used to restore some visual function, and non-invasive neuromodulation could be added to further improve vision. There are many possibilities of combining different approaches that will depend on the particular scenario. At this point, combining different methods for visual neurorestoration is theoretical, but various groups (including our own) are working toward multimodality approaches to visual neurorestoration.

Any visual restorative therapy faces significant barriers to implementation, from the early design of novel therapeutics to the continued use of established therapies. The first major barrier is securing financing, which is often dependent on federal research grants that are very competitive and certainly insufficient to develop a novel treatment for human use, invariably necessitating industry partnership. Industry involvement comes with its own ethical issues, and it shifts the focus away from efficacy to marketability [265]. Furthermore, industry has the ability to grant or deny rights of use for expensive technologies, which can stop the development of therapies in their tracks [266]. If therapies can secure financing, FDA approval is the next major barrier, which imposes strict guidelines and has high costs in terms of time and effort on the part of researchers and clinicians. In the case of the Argus II, a humanitarian device exemption (HDE) was obtained [267], which would likely be true for any future implantable device coming to market. An HDE designation can be helpful, as it abrogates the need for extensive clinical trials and is available for diseases with small numbers of patients; however, it can also be harmful to the development of these devices, as it precludes rigorous testing for safety and efficacy [268]. As neurosurgeons, we are familiar with the decline in deep brain stimulation for obsessive compulsive disorder after this therapy received an HDE. This was due to decreased funding from industry for clinical trials, and insurance companies frequently deny insurance authorization [269]. Similarly, Cortigent, Inc. discontinued manufacture of the Argus II after an HDE designation was obtained due to the high costs of production and a limited number of patients. One of the sad realities for most of the therapies discussed in this manuscript is that they are expensive and can only be used in small patient populations. This fact makes all of these therapies susceptible to financial pressures. Legislation like the Orphan Drug Act was designed to stimulate the development of treatments for rare diseases and has been helpful in incentivizing companies to develop therapies for rare causes of blindness, such as in the case of Luxturna [270]; however, there are also problems with orphan drugs, such as high costs and less attention to safety and efficacy [271].

There are also myriad ethical considerations for all the visual neurorestoration strategies discussed in this manuscript. The most obvious of these ethical issues is the use of embryonic stem cells. There are, of course, the moral and societal considerations as to whether an embryo is a living human being; furthermore, there are risks to donors and recipients [272]. The use of autologous stem cells bypasses some of these ethical considerations, but there is still some ethical concern about the use of these cells. Namely, reprogramming human cells has the potential for malignant transformation; there is a concern for privacy and protection of personal health information; and there is a concern that iPSCs could be used to generate human embryos [273]. Along those same lines, there are concerns about genetic engineering. Specifically, there are concerns with respect to off-target effects such as malignant transformation, germline incorporation of mutations that could be passed to offspring, and difficulty in obtaining informed consent when long-term effects are unknown [274]. With regard to whole-eye transplant, there are some ethical concerns with regard to protecting the identity and dignity of donors and recipients, the potential for inequity in the selection of recipients, and the potential for harm caused by immunosuppression [275]. Finally, as these are all very expensive therapies, there are ethical considerations with regard to healthcare disparities and providing access to care [276].

Despite the practical, financial, and ethical limitations in the current technologies for visual neurorestoration, there have been significant advances in regenerative therapies and neuromodulation that have required considerable effort and innovation. We are in an era in which these technologies will continue to accelerate toward the goal of visual neurorestoration in humans. In order to harness the unique capabilities of all strategies in visual neurorestoration, multidisciplinary teams of ophthalmologists, bioengineers, neurosurgeons, plastic/reconstructive surgeons, and neuroscientists will need to be assembled. Projects like ARPA-H THEA are one such avenue for this type of collaboration; however, there needs to be more collaboration to facilitate growth in approaches beyond eye transplantation. Future efforts will need to be made to improve legislation regarding orphan drugs and devices with HDEs so that clinical trials can be performed with the level of rigor necessary to find efficacious therapies and so that these therapies can obtain insurance coverage and are affordable enough to provide access to patients from different socioeconomic backgrounds and regions.

## 7. Limitations

One of the major limitations in this article is that it draws from many preliminary case reports, case series, and phase 1 and phase 2 studies. As such, these studies often involve a small number of patients, are not randomized, and are not blinded. Furthermore, having a sham condition for many of these procedures would not be feasible or ethical. Most of the studies have the internal control of one treated eye and one untreated eye for the participants, but many studies do not include controls of untreated individuals. Therefore, the results of many of these studies are subject to biases, which are, in turn, propagated in this review article. Where applicable, we attempted to focus on the objective data from the studies, such as BCVA; however, even these objective measures could be subject to confounding and the placebo effect.

Another limitation of this review article is that there are multiple sources of bias. First, most of the authors of this manuscript are neurosurgeons with a primary interest in invasive and non-invasive neuromodulation, as well as brain–computer interfaces. Therefore, the sections dealing with neuromodulation may be over-represented relative to their prevalence as treatment options in ocular diseases. Second, the senior ophthalmologist in this study has a research interest in optic nerve regeneration after injury and is an investigator in the ARPA-H whole-eye transplant project, which has influenced our team’s opinions about the importance of eye transplantation. Furthermore, the ophthalmology department at the University of Southern California has had critical involvement in cellular therapies and the development and testing of the Argus II. Given our relationship with other investigators at our institution, this could influence the emphasis we place on cellular therapies and retinal neuroprosthetics. These limitations, however, have contributed to our own views on the field of visual neurorestoration and have fostered a sense of excitement that the authors of this manuscript hope to share with the readers.

## 8. Conclusion

The visual pathway is an incredibly complex and sensitive system, and replacing damaged components of this system requires complex solutions. There have been considerable recent innovations in stem cell biology, genetic engineering, optogenetics, tissue transplant, neuroprosthetics, and non-invasive neuromodulation. Despite these innovations, there is considerable work to be done to help patients see again, which requires a concerted, multidisciplinary effort involving scientists and clinicians, as well as a considerable investment from industry and society.

## 9. Methods

As this is an expert review, it relies on a considerable body of background information that was achieved through individual searches on PubMed, Google Scholar, and our institution’s library (USC Norris Library) to find the necessary studies and texts. Since the emphasis of this manuscript is on human clinical trials, we obtained the studies discussed herein through targeted searches on PubMed. Below are the search terms that we used to identify studies for this manuscript for the cellular therapeutics, genetic engineering, and optogenetics sections, but certain studies were omitted if they were not appropriate or added. We included original articles that discussed the results of clinical trials and excluded related studies that discussed the methods of clinical trials without discussing the primary clinical outcomes of the trial. We only searched for articles in the English language. The studies included ranged from November 2009 to September 2025. The search terms applied were used to identify clinical trials. As for quality, all studies involved appeared in peer-reviewed journals that are PubMed-indexed. The minimum number of patients was one human patient. Although the gold standard in human studies is a randomized, sham/placebo-controlled trial with blinding, the studies involved are mostly phase I and phase II studies in which randomization, blinding, and placebo control are not feasible or ethical at this time. Therefore, we assessed the articles for quality according to whether the article included objective data such as visual acuity metrics, optical coherence tomography, and fundoscopy. We included all studies identified in the following searches.

Cellular therapeutics:

(“Stem Cells”[TIAB] OR “Embryonic Stem Cells”[TIAB] OR “Induced Pluripotent Stem Cells”[TIAB] OR “iPSC”[TIAB] OR “hESC”[TIAB] OR “mesenchymal”[TIAB]) AND (“Retinal Disease”[TIAB] OR “Inherited Retinal Diseases”[TIAB] OR “Macular Degeneration”[TIAB] OR “Age-Related Macular Degeneration”[TIAB] OR “Diabetic Retinopathy”[TIAB] OR “Glaucoma”[TIAB] OR “Stargardt Disease”[TIAB] OR “Achromatopsia”[TIAB] OR “Bietti Crystalline Dystrophy”[TIAB] OR “Choroideremia”[TIAB] OR “Usher Syndrome”[TIAB] OR “X-linked retinitis pigmentosa”[TIAB] OR “X-linked retinoschisis”[TIAB] OR “Leber Hereditary Optic Neuropathy”[TIAB]) AND (“Randomized Controlled Trial”[PTYP] OR “Clinical Trial”[PTYP] OR “Clinical Trial, Phase I”[PTYP] OR “Clinical Trial, Phase II”[PTYP] OR “Clinical Trial, Phase III”[PTYP]) NOT “Review”[PTYP]

Gene therapy:

(“Gene therapy”[TIAB]) AND (“Retinal Disease”[TIAB] OR “Inherited Retinal Diseases”[TIAB] OR “Macular Degeneration”[TIAB] OR “Age-Related Macular Degeneration”[TIAB] OR “Diabetic Retinopathy”[TIAB] OR “Glaucoma”[TIAB] OR “Stargardt Disease”[TIAB] OR “Achromatopsia”[TIAB] OR “Bietti Crystalline Dystrophy”[TIAB] OR “Choroideremia”[TIAB] OR “Usher Syndrome”[TIAB] OR “X-linked retinitis pigmentosa”[TIAB] OR “X-linked retinoschisis”[TIAB] OR “Leber Hereditary Optic Neuropathy”[TIAB]) AND (“Randomized Controlled Trial”[PTYP] OR “Clinical Trial”[PTYP] OR “Clinical Trial, Phase I”[PTYP] OR “Clinical Trial, Phase II”[PTYP] OR “Clinical Trial, Phase III”[PTYP]) NOT “Review”[PTYP]

Where applicable, we discussed some ongoing clinical trials that have not yet published results; however, these studies are, for the most part, out of the scope of this review. We did not perform a generalized search for the “Whole-Eye Transplantation” section, the “Optogenetics” section, the “Neuroprosthetics” section, or the “Neuromodulation” section, as there were very small numbers of studies.

For the generation of figures, BioRender.com was used. Jonathon Cavaleri has a personal software license and used BioRender.com to generate figures. For all other figures that were sourced from other sources, permission was obtained for reproduction and submitted to the journal.

## Figures and Tables

**Figure 1 brainsci-15-01170-f001:**
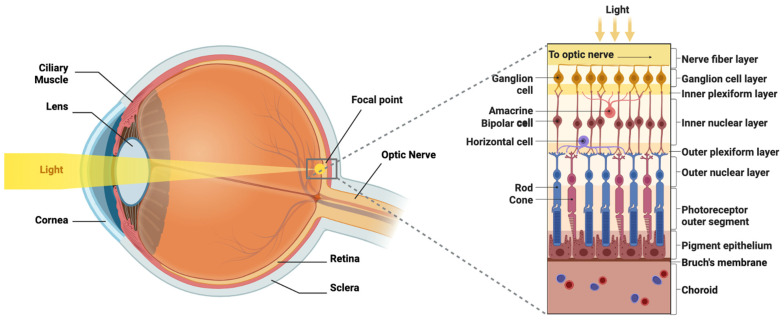
Visual anatomy. The **left side** depicts a cross-section through the eye and optic nerve in a sagittal orientation, with a beam of light being refracted by the cornea and lens and focused onto the retina. The **right side** depicts a zoomed-in cross-section of the retinal cell layers with the vitreal surface at the top and the choroidal surface at the bottom. Image generated with BioRender.com with authorization.

**Figure 2 brainsci-15-01170-f002:**
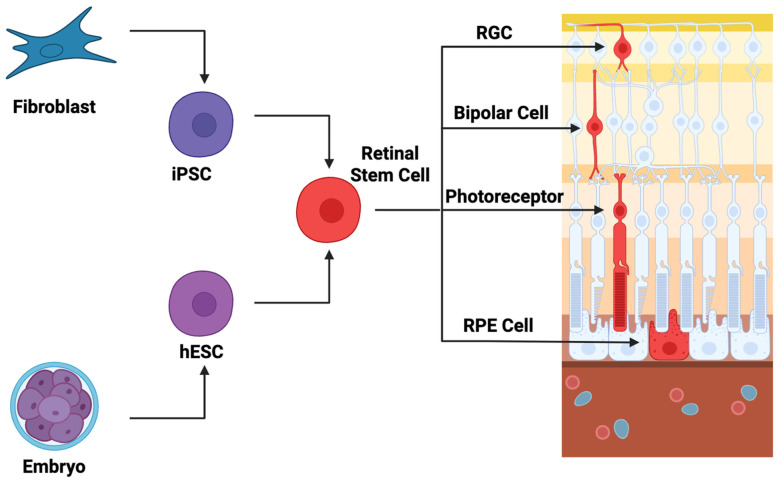
Stem cell therapies. The **left side** shows the two possible cellular origins for retinal stem cells: induced pluripotent stem cells (iPSCs) and human embryonic stem cells (hESCs). The **right side** shows the possible cell types into which the retinal stem cells can be differentiated: retinal ganglion cells (RGCs), bipolar cells, photoreceptors, and retinal pigment epithelial (RPE) cells. Image generated with BioRender.com with authorization.

**Figure 3 brainsci-15-01170-f003:**
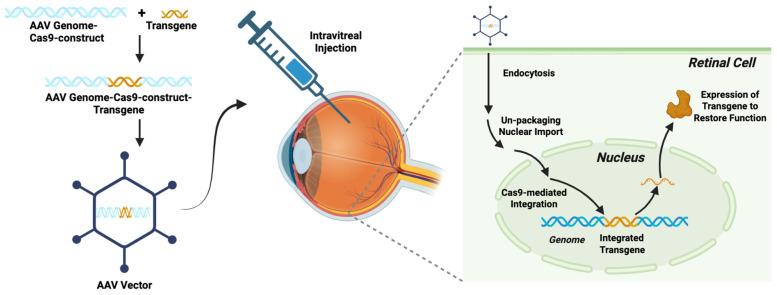
Gene therapies. The **left side** of the figure shows the insertion of the transgene into the adenovirus-associated virus (AAV) genome and packaging into the viral capsid with intravitreal injection. The **right side** depicts viral entry into the target retinal cell, transgenic integration via CRISPR-Cas9-mediated gene editing, and then gene expression of the transgene of interest. Image generated with BioRender.com with authorization.

**Figure 4 brainsci-15-01170-f004:**
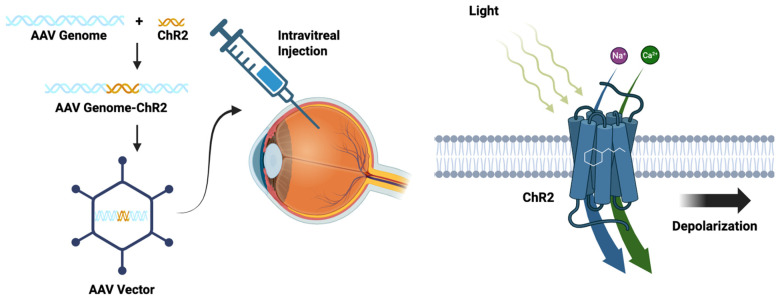
Optogenetic therapies. The **left side** of the diagram depicts insertion of the optogenetic transgene (ChR2) into the adenovirus-associated virus (AAV) genome and packaging into the viral capsid with intravitreal injection. The **right side** shows the mechanism of action of ChR2, which is a photon-gated cation channel that acts to depolarize the cell, leading to action potential. Image generated with BioRender.com with authorization.

**Figure 5 brainsci-15-01170-f005:**
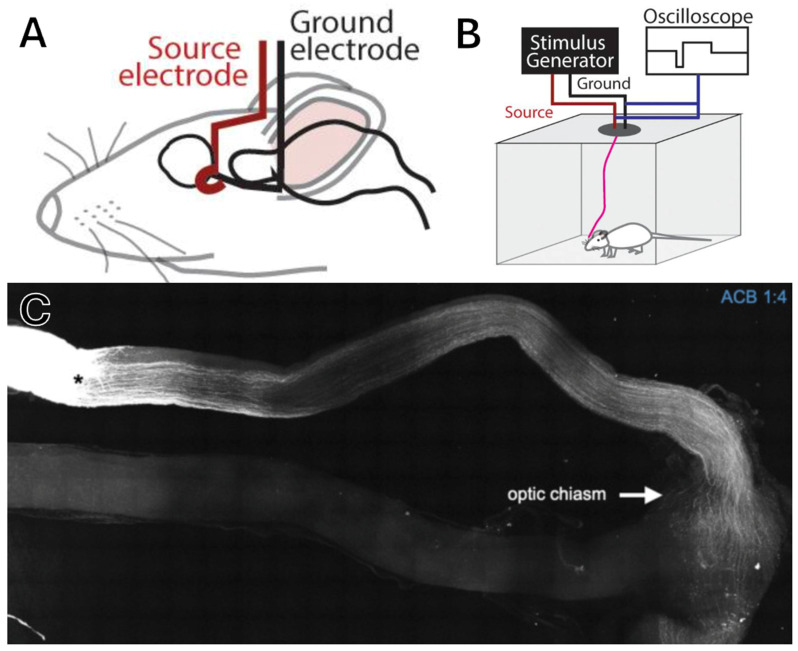
Electric field for axon regeneration after crush injury (modified from Kim et al., with permission for reproduction). (**A**) Illustration of optic nerve source electrode and chiasmatic ground electrode. (**B**) Illustration of experimental setup in which chronic stimulation can be applied to an awake rat and potentials can be recorded. (**C**) Cholera toxin B-labeled RGC axons are noted to grow past the crush site (*) down the length of the optic nerve and through the optic chiasm when asymmetric charge-balanced waveform fields are passed along the length of the nerve [185].

**Figure 6 brainsci-15-01170-f006:**
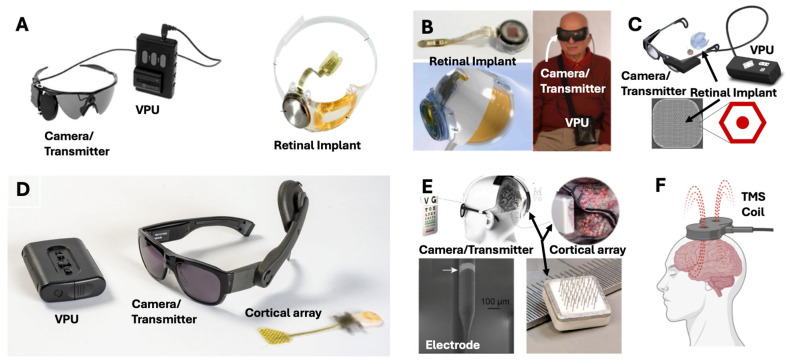
Visual neuroprosthetics and neuromodulation. (**A**) Argus II device (modified from Farvardin et al., with permission for reproduction) [190], (**B**) IRIS-II device (modified from Muqit et al., with permission for reproduction) [191], (**C**) PRIMA device (modified from Muqit et al., with permission for reproduction) [192], (**D**) Orion device (reproduced with permission from Cortigent, Inc.) [193], (**E**) Gennaris device (modified from Rosenfeld et al., with permission for reproduction) [194], (**F**) transcranial magnetic stimulation (TMS) (image generated with BioRender.com with authorization).

**Table 1 brainsci-15-01170-t001:** Summary of the various modalities of visual neurorestoration, conditions treated, and visual outcomes. Abbreviations: Human embryonic stem cells (hESCs), induced pluripotent stem cells (iPSCs), age-related macular degeneration (AMD), best corrected visual acuity (BCVA), adeno-associated virus (AAV) retinitis pigmentosa (RP), Leber Hereditary Optic Neuropathy (LHON), Intelligent Retinal Implant System (IRIS), Photovoltaic Retinal Implant (PRIMA), Artificial Vision by Direct Optic Nerve Electrical stimulation (AV-DONE), lateral geniculate nucleus (LGN), repetitive transcranial magnetic stimulation (rTMS), transcranial direct current stimulation (tDCS), transcranial alternating current stimulation (tDCS), transcranial random noise stimulation (tRNS).

Modality	Examples	Conditions Treated	Visual Outcome
Stem Cell Therapy	hESCs, iPSCs, mesenchymal	AMD, Stargardt’s Disease	BCVA ≥ 0–20 words
Gene Therapy	AAV vectors delivering transgenes	Achromatopsia, Bietti’s crystalline dystrophy, choroideremia, autosomal RP, X-linked RP, Stargardt disease, Usher syndrome, X-linked retinoschisis, and LHON, AMD, diabetic retinopathy, glaucoma	BCVA ≥ 0–20 words
Optogenetics	AAV vectors delivering ChrimsonR, ChR2, ChronosFP, MCO1	RP, Stargardt’s disease	Identification of objects in a blind patient
Whole-Eye Transplant	Eye/partial face transplant	Severe burn injury	No visual restoration
Retinal Neuroprosthetics	Argus II, IRIS II, PRIMA	RP, AMD	Object/picture recognition, motion detection
Optic Nerve Neuroprosthetics	AV-DONE	RP	Visual percepts elicited in blind patients
Thalamic Neuroprosthetics	Micro-electrode stimulation of LGN in non-human primates	N/A	Visual percepts elicited
Cortical Neuroprosthetics	Orion, Gennaris, Utah Array	Complete blindness	Visual percepts elicited, shapes recognized in blind patients
Non-Invasive Neuromodulation	rTMS, tDCS, tACS, tRNS	Amblyopia, hemianopsia post-stroke, normal vision	Improvements in contrast sensitivity, motion perception, visual field function, crowding in peripheral vision, and neglect

## Data Availability

No new data were created or analyzed in this study.

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
