# Peer review of "Visual Neurorestoration: An Expert Review of Current Strategies for Restoring Vision in Humans"

_brainsci, 2025, doi:10.3390/brainsci15111170_

Round 1

Reviewer 1 Report

Comments and Suggestions for Authors

  1. I noticed a lot of numerical placeholders (like "17" or "68" repeated across pages) that seem like OCR errors or unfinished sections. These make it tricky to follow your arguments. Could you replace these with the intended content or remove them to ensure the text flows smoothly?

  2. The manuscript covers an impressive range of topics, from eye anatomy to cutting-edge therapies. However, the transitions between sections feel a bit abrupt. Adding clear section headings or brief introductory sentences could help guide readers through your thought process more seamlessly.

  3. Your discussion of therapies like RKG-314 and the THEIA project is fascinating! To make it even more impactful, could you clarify how these compare to current treatments in terms of efficacy or practicality? A table summarizing key therapies and their clinical trial stages might help readers grasp their potential.

  4. The references you’ve included are solid and relevant, but the field of visual neurorestoration is moving fast. Adding a few more recent studies (post-2020) on topics like CRISPR-based gene editing or optogenetics could make your review feel even more current and comprehensive.

  5. The description of Figure 1 is helpful, but ensuring all figures are clearly referenced and described in the text would improve clarity. Also, could you provide more details on the authorization process for images generated with tools like Fluorender? This would address any ethical questions about image use.

  6. There are some typographical errors (e.g., "Ocpptal Cortex" or "FEEK REVIEW") and truncated sentences that disrupt the flow. A thorough proofreading, perhaps with the help of a professional editor, would polish the manuscript and make it more engaging for readers.

Reviewer 2 Report

Comments and Suggestions for Authors

Can the observed enhancements in certain trials be ascribed to variables other than the intervention (e.g., placebo effects)? What measures were implemented to regulate these factors?

Were the figures altered or annotated to highlight certain results or methods addressed in the manuscript? Otherwise, how do they uniquely add to the narrative?

What criteria were employed to select the papers and clinical trials for inclusion in this review? Were there particular inclusion and exclusion criteria, and if so, what were they?

What ethical considerations are involved with gene editing or complete eye transplantation, and how may they affect the translation of these medicines into clinical practice?

What precise measures are required to promote interdisciplinary collaboration (e.g., neurosurgery, ophthalmology, bioengineering) to enhance visual neurorestoration? Do any current projects or consortia address these requirements?

Were any actions implemented to evaluate or alleviate any biases in the examined studies? How do the writers rationalize the potential impact of bias on their conclusions?

What was the reason for not doing a meta-analysis to quantitatively assess the efficacy of various therapies? Would this analysis not reinforce the conclusions?

What precise measurements or standards were employed to categorize gains as "modest"? Are there established metrics for assessing vision restoration outcomes?

What are the logistical, ethical, and regulatory obstacles to integrating these medicines, and how could they be surmounted?

What are the long-term results of entire eye transplantation regarding graft survival and functional vision? Is there any data available beyond the one-year follow-up referenced?

How can the authors reconcile the elevated failure rates or restricted efficacy of some therapies (e.g., cortical implants) with their hopeful representation in the discussion?

Reviewer 3 Report

Comments and Suggestions for Authors

Dear Authors,

I have read with interest your manuscript entitled "Strategies in Visual Neurorestoration".

At the present stage, I find the manuscript methodologically weak, and with a very significant risk of bias. In particular, I note the following shortcomings:

Title: the title is unsuitable, and should be revised so that it is clear that the article is a review. As no canonical protocol (e.g., PRISMA systematic, PRISMA scoping) has been followed, it would be preferrable if the type of review could be qualified (e.g., "Expert review").

Abstract: similarly, the fact that the article is a review, and the type of review, should be in good evidence in the abstract.

Introduction (and, again, title and abstract): once read the introduction, it is still to me very unclear what the specific aims and objectives of this review are (the "scientific question"). If we look at well-posed review structures such as, for example, again those proposed in the different variants of the PRISMA statement, we note how reviews are expected to derive from a research question and, indeed, the nature of this research question determines the nature of the review (scoping vs systematic vs meta-analysis etc.). Here I do not find any explicit research question, hence I have no reading key for the rest of the review. Many frameworks are available for a solid formulation of research questions (e. g.  PICO, PCC, SPICE) and a suitable one should be used and referenced. This should most definitely be addressed in the introduction, but also be clear in the abstract and, if at all possible, inform a good title.

Please note that this is not merely a matter of form. At present, the paper does not offer any indication on an objective search strategy for the literature reviewed, selection and exclusion of sources, selection and exclusion of literature items, analysis of the quality of the items, data extraction, etc. In this sense, it appears to follow the overall structure of an "expert review" which, ultimately, is an opinion paper supported by literature. For this very reason, the presence of a specific research question and a precise justification are essential, otherwise, in addition to raising serious questions of usefulness, an expert review raises subjectivity and bias issues (which unavoidable) which cannot be evaluated.

Section 4.1 (with notes relevant to the overall paper structure): I note on line 521 the statement "In 2019, production of the Argus II was halted by the manufacturer, but production was set to be resumed by Cortigent Inc. in 2023". We are in 2025, and a statement on what will happen in 2023 is indeed at least two years old. This is extremely worrying. A simple check of the Cortigent site advises that production has been halted permanently, giving very strong core reasons. In practice, in the presence of a corporate website for a key player in the sector, the manuscript mentions an outdated article on an IEEE outreach magazine.

This is a very serious problem under many fronts. Firstly, it indicates that the literature referenced is not up to date and not authoritative. Very importantly, it misses out core limitations of the technology that the Cortigent website clearly outlines, that should have been identified through a trivial search (the website of the very company being referenced!). In turn, this raises doubts on the reliability of the literature search methodology, given that this methodology is not provided, as per my notes above.

Similarly, and very worryingly, the very same section 4.1 moves on from Cortigent by discussing devices by Pixium Vision. However, these are discussed on the basis of papers published a couple of years ago, yet on older studies. Again, a simple check on the corporate website shows that Pixium Vision no longer exists, and has now been absorbed by LaScience SAS. This is again a non-trivial piece of information missed by omitting a trivial search of a relevant website, strengthening the doubts and reservations on the literature search methods.

Please note that the two above are merely examples. The core problem does not sit with what the manuscript says about Cortigent or Pixium but, rather, with the fact that the literature search methodology is not explored, there is no research question, and therefore the work is open to bias.

Section 6 (future directions) appears unduly scant with respect to the other sections of this paper. Most sections describe research that is undergoing very active development, with techniques being explored, companies being set up, other companies being bought, etc. Yet, the future perspectives of this all are dismissed in five lines (687-691) with purely subjective considerations. This section should instead match the many threads left open by the previous sections, to draw a grounded and motivated picture that emerges from them. Again, without a specific research question this section cannot be evaluated in detail though, most certainly, five lines that, effectively, dismiss the totality of the technologies on the landscape, followed by the mention,  in prominent evidence in lines 708-712, of a specific project that involves some authors, is highly suggestive of significant bias. While indeed the authors may be absolutely correct in their considerations, the conclusions require far more grounding in the literature evidence.

There are formal errors. For example,

  • the acronym "RGC" is first used on line 154, but only defined later, in line 180.
  • lines 161-185, 240-247, 306-313 are not formatted consistently with the rest of the paper (centred with no margin, instead of justified with appropriate margin).

Round 2

Reviewer 1 Report

Comments and Suggestions for Authors

The authors have addressed all concerns, revising the manuscript to improve clarity, update references, correct formatting issues, and provide more comprehensive explanations of the therapies discussed.

Author Response

Dear Reviewer,

Thank you so much for taking the time to review our manuscript and for providing us with your thoughtful insights. Your suggestions made for a much stronger review article. 

Thank you,

Jonathon Cavaleri, MD

Reviewer 2 Report

Comments and Suggestions for Authors

The author responded to my comments.

Author Response

Dear reviewer,

We would like to thank you for taking the time to review our manuscript. Thanks to your insights, our manuscript is much stronger. 

Thank you,

Jonathon Cavaleri, MD